# Biohacking Nerve Repair: Novel Biomaterials, Local Drug Delivery, Electrical Stimulation, and Allografts to Aid Surgical Repair

**DOI:** 10.3390/bioengineering11080776

**Published:** 2024-07-31

**Authors:** Jordan R. Crabtree, Chilando M. Mulenga, Khoa Tran, Konstantin Feinberg, J. Paul Santerre, Gregory H. Borschel

**Affiliations:** 1Division of Plastic Surgery, Department of Surgery, Indiana University School of Medicine, Indianapolis, IN 46202, USA; 2Institute of Biomedical Engineering, University of Toronto, 164 College St Room 407, Toronto, ON M5S 3G9, Canada; 3Department of Ophthalmology, Indiana University School of Medicine, Indianapolis, IN 46202, USA

**Keywords:** nerve regeneration, peripheral nerve injury, nerve conduit, electrical stimulation, autograft, allograft

## Abstract

The regenerative capacity of the peripheral nervous system is limited, and peripheral nerve injuries often result in incomplete healing and poor outcomes even after repair. Transection injuries that induce a nerve gap necessitate microsurgical intervention; however, even the current gold standard of repair, autologous nerve graft, frequently results in poor functional recovery. Several interventions have been developed to augment the surgical repair of peripheral nerves, and the application of functional biomaterials, local delivery of bioactive substances, electrical stimulation, and allografts are among the most promising approaches to enhance innate healing across a nerve gap. Biocompatible polymers with optimized degradation rates, topographic features, and other functions provided by their composition have been incorporated into novel nerve conduits (NCs). Many of these allow for the delivery of drugs, neurotrophic factors, and whole cells locally to nerve repair sites, mitigating adverse effects that limit their systemic use. The electrical stimulation of repaired nerves in the perioperative period has shown benefits to healing and recovery in human trials, and novel biomaterials to enhance these effects show promise in preclinical models. The use of acellular nerve allografts (ANAs) circumvents the morbidity of donor nerve harvest necessitated by the use of autografts, and improvements in tissue-processing techniques may allow for more readily available and cost-effective options. Each of these interventions aid in neural regeneration after repair when applied independently, and their differing forms, benefits, and methods of application present ample opportunity for synergistic effects when applied in combination.

## 1. Introduction

In the opening paragraph of a review covering peripheral nerve repair published in 1944, Weiss included the following comment on arterial sleeve cuffing, an emerging technique trending toward clinical adoption at the time:

“*But further application of the lessons thus learned gives promise of even more substantial improvements, and sleeve splicing may eventually be superseded by some other, more meritorious procedure incorporating its experiences. The emphasis lies more on the principle than on the current form of its application…*”[1]

Relative to the technique described—namely, connecting severed nerve ends with donor infant aorta without suture—methods of repair for peripheral nerve injury (PNI) have advanced greatly, particularly concerning the available materials employed to guide axonal regeneration. Arterial sleeve cuffing is no longer applied in nerve repair, and the prediction made by Weiss appears true 80 years later, as the introduction and widespread adoption of biomaterials into clinical use and research has allowed for recent advancements and even greater potential for success in the future of nerve reconstruction after injury.

Unfortunately, the management of peripheral nervous system injuries remains challenging, with approximately one-third of patients with PNI exhibiting incomplete recovery and poor outcomes such as chronic pain, deficits in motor and sensory function, and muscle atrophy [2]. These outcomes are even worse in cases where a large gap exists between the proximal and distal stumps to be reconnected, and the use of an interposed graft or other nerve substitute is required to achieve a tension-free repair.

In mammalians, unlike in cold-blooded organisms, the nerves of the central nervous system (CNS) fail to regenerate after injury. In contrast, the nerves of the peripheral nervous system (PNS) retain their regenerative capacity, which mostly relies on the activity of the peripheral glial cells—the axon-ensheathing myelinating and nonmyelinating Schwann cells (SCs) [3].

Upon transection of a peripheral nerve, axons become disconnected from their cell bodies, triggering Wallerian degeneration distal to the injury. This begins with the breakdown of the myelin sheath through SC autophagy and macrophage recruitment, with subsequent denervation of the SCs associated with it. Once denervated, changes in SC gene expression lead to their transdifferentiation to a phenotype better equipped for supporting axonal growth, regeneration, and eventual reinnervation of target tissues [3,4,5]. Once transdifferentiated, SCs can perform repair functions, including the clearance of pathologic myelin to create a path for new axons, regulation of nutrient exchange with regenerating axons, and the promotion of new myelin production, among others [3,6,7]. A crucial aspect of this phenotypic shift and subsequent nerve healing is the upregulation of SC genes responsible for the secretion of several neurotrophic and chemotactic factors influencing cellular growth and migration [6].

In order for healing to take place across a nerve gap, numerous cells from both nerve stumps must be able to traverse the deficit in a well-defined order. This process begins with the secretion of factors and extracellular matrix (ECM) precursors from both transected nerve stumps, which solidify to a matrix of ECM cables, providing a bridge for SC crossing. SCs play a key role in guiding the remainder of axonal regeneration, as SC migration along the cables of this matrix is closely followed by the migration of endothelial cells and fibroblasts from both nerve stumps, and then axons from the proximal nerve stump, which are myelinated soon after. These processes were first described by Williams et al. in 1983 [8] and have since been delineated to five phases: (i) the fluid phase; (ii) the matrix phase; (iii) the cellular migration phase; (iv) the axonal phase; and (v) the myelination phase [8,9,10,11]. However, although possible, the process of neuronal regeneration in the PNS is often very slow and can be affected by several pathological processes associated with trauma, such as a large gap between the proximal and distal segments of a truncated nerve, local inflammation, ischemia, or background disorders, such as diabetes or congenital conditions [12,13].

Several therapies have been developed to affect the speed and quality of this natural recovery process with documented success in clinical application, and many more can be found in all phases of development. In general, this progression consists of first showing in vitro benefits through application to isolated and cultured cells. Next, an animal model is employed, most commonly the rat sciatic nerve due to its accessibility, size, and ease of functional recovery monitoring. After decades of confirmation by many groups, the therapy may be investigated in human trials of increasing rigor, size, and diversity, before consideration by regulatory authorities for its use outside of experimental platforms.

This review article provides a three-pronged comprehensive examination of (i) therapies that have undergone rigorous human clinical trials and are currently employed in clinical practice, (ii) therapies with characteristics that will allow particular ease of translation into clinical practice, and (iii) recently described, novel therapies that directly complement those already in use.

### History of Biomaterials in Nerve Repair

Less than 100 years passed between the development of the first plastic material in the 1860s and the first described use of a plastic polymer implanted in the human body in 1939, when a fibrosis-inducing wrap was used to mitigate the expansion of an arterial aneurysm [14,15]. Since this first described application, the widespread adoption of both natural and synthetic polymeric materials as biomaterials has revolutionized the field of surgery.

A biomaterial is defined as “a material designed to take a form that can direct, through interactions with living systems, the course of any therapeutic or diagnostic procedure” [16]. These materials have widely varying functions, strengths, and limitations owing to their diverse chemical formulations and fabrication methods. For the purpose of peripheral nerve repair, biomaterials have most commonly been formed into nerve conduits (NCs), which are tubular supports fashioned around areas where neural healing is to be encouraged. More specifically for this review, this latter definition will be the context in which biomaterials will be discussed. Interventions involving the manipulation of living or recently living tissues for bridging a deficit will be referred to generally as “grafts”. Finally, “nerve guide” will be used as an umbrella term to include both NC and graft approaches, though many groups use these terms interchangeably in their descriptions.

The origin of biomaterial-based nerve guides is often cited as the use of decalcified bone tubes to facilitate nerve repair by Gluck in 1880 [17]. For many years following, research focused largely on the use of autogenous tissues as nerve guides, such as blood vessels (1891) [18], fascia (1915) [19], and skeletal muscle (1940) [20], though intermittently, researchers had leveraged nonbiologic materials including magnesium (1900) [21], gelatin (1901) [22], and galalith (1915) [23]. In 1946, Weiss and Taylor bridged nerve gaps in animal models with collagen sleeves, a material still in use today, though cited rapid resorption of the material as a reason for only rare success [24]. As a direct result of these early studies and the long history of poor outcomes in nerve repair, decades of research have explored what properties make up the ideal nerve guide, with novel biomaterials accounting for recent success in addressing the limitations delineated from earlier attempts. The five phases of neural regeneration are illustrated in Figure 1A, along with interventions currently in use or under study to enhance these phases (Figure 1B–L).

## 2. Nerve Conduits

### 2.1. Structural Properties

NCs have been studied in a range of forms, and broad structural categories have been described as generations: first-generation conduits are hollow, tubular structures used for support and as a barrier to the environment; second-generation conduits are resorbable and biocompatible with specific wall structures to guide axonal growth topographically; and third-generation conduits incorporate other bioactive functions, such as luminal fillers, drugs to enhance regeneration, or cellular or extracellular components [36]. Within these large categories, NC structures can be further distinguished by their relative ability to degrade within the body, degree of porosity or permeability, and the specific topographic cues of their walls or intraluminal contents. The optimal characteristics among these have shifted greatly since early NC designs and continue to evolve as research progresses. Here, we offer a summary of how the field’s preferences have evolved, from the early concepts to where they currently lie.

#### 2.1.1. Degradation

Nerve conduits in their simplest form exert their effects by directing elongating nerve fibers along a deficit to be bridged. This was the primary goal of early synthetic conduits composed of relatively inert, relatively poor-degrading materials such as silicone, which Lundborg et al. described in 1979 as the “chamber principle” of isolating a healing nerve [37,38]. It was theorized that enclosing the gap between two nerve ends could require fewer sutures, mitigate excess surgical trauma, prevent the invasion of the healing nerve deficit by fibrous scar tissue, and allow luminal accumulation of neurotrophic and neurite-promoting factors [9]. However, the use of a material that did not degrade within the body occasionally necessitated reoperation for removal due to delayed tissue reactions and fibrosis [37,38,39]. At the time of this early conduit’s use, the author noted that a second surgery was preferable to the use of material that would degrade and could induce inflammation and fibrosis, or interfere with environmental factors involved in healing [39].

Soon after, this view shifted. Nerve conduits are now designed to be biodegradable rather than inert to avoid the risks of retained foreign material at the repair site, as recognized by Merle et al. in 1989 [36,40,41]. At the same time, this degradation should not take place fast enough to trigger an inflammatory response or jeopardize the strength, shape, and axon-guiding ability of the NC [42]. The flexibility afforded by using biodegradable polymers in modern NCs addresses these concerns, with the added benefit of degradation often being tied to several strategies shown to aid in nerve healing, such as drug, growth factor, or cell delivery. Additionally, advancements toward the generation of functional materials, that can degrade in vivo, have allowed for the application of physical stimulation in innovative strategies, such as using biodegradable electrodes and nerve stimulators, discussed later in this review.

#### 2.1.2. Topographic Features

Stemming from observations in 1912 that cells in culture moved along and adapted to the form of spider webs [43], “contact guidance” is a term used to describe the propensity of cells to adjust their orientation to align with groove-like patterns when growing on them [44]. Many NC fabrication techniques leverage this principle with topographical guidance cues, to enhance cellular alignment for migration and provide directionality to axonal tract regeneration. These include longitudinal topography introduced within conduit walls through aligned channels and/or grooves [29,45,46,47], or the generation of aligned fibers [48,49]. Additionally, aligned structural supports extending across the center of the conduit such as microchannels [50] or filaments of defined sizes and compositions [31,51,52] have been employed to encourage migration along their length. The function of these microfeatures may be enhanced by biochemical cues, such as those associated with extracellular matrix (ECM) proteins, using these approaches to promote migration [50]. The manipulation of these ECM proteins is also frequently cited as a method to tailor the “roughness” of a conduit’s surface to better allow native SC attachment and migration, or the survival of cells delivered with the NC [50,53,54].

The inclusion of hydrogels within some NC designs represents a distinct but related approach to topographic guidance. Hydrogels are composed of hyperhydrophilic polymer chains that exhibit a diverse set of structural characteristics, defined in part by their crosslinking degree and nature [55]. Through modulation of these crosslinks, hydrogels can be prepared as liquids and injected at a site or into a conduit prior to solidifying, via specific mechanisms such as changing temperature, pH, or time-dependent chemical gelation [55,56]. As a result, hydrogels often serve as a medium in which interventions such as drugs or cells are co-delivered with the NC materials, thereby also providing structural and topographic support within the lumen of an NC. Examples include the incorporation of ECM proteins into the gel matrix to be used as bioinstructive cues [56].

More recent advancements in NC fabrication techniques continue to allow for greater control over the topographic characteristics of all types of NCs. One example can be found in the application of shape memory nanofibers, applied in multichannel conduits of aligned and random orientations by Wang et al. in 2020 [30]. Additionally, the widespread advancement and adoption of 3D printing technology over the past decade has permitted a level of structural customization in nerve guides, that are not attainable by many other, more established NC production techniques. These include the fabrication of bifurcated and irregular conduits, reverse-engineered from 3D anatomical scans by Johnson et al. in 2015 [28], and many other potential functional integrations to allow for personalized therapies within PNI, as recently reviewed by Liu et al. [57].

#### 2.1.3. Porous Structure and Permeability

Another early quality discovered to be vital in developing NCs was permeability and its associated porosity—defined as the ratio of the volume of interconnected void space divided by the total volume of a material [58]. This was exemplified by the use of an NC composed of expanded polytetrafluoroethylene (ePTFE) in 1998, which was designed to interact differentially with proteins in comparison to silicone, despite being relatively bio-stable like the latter material. Specifically, ePTFE allowed nutrient exchange as a result of its interconnected porous walls [59]. Other early notable porous NCs include some composed of biodegradable polymers such as polyglycolic acid (PGA) [60,61]. Permeability remains a key characteristic under study within nerve conduits to this day, with efforts to describe the ideal characteristics for neural regeneration in porosity, pore size, connectivity, uniformity, and three-dimensional morphology still ongoing [62].

Advancements in biomaterial fabrication processes have allowed conduits of different porous structures to be readily made, and several tissue-regeneration outcomes have been attributed to varying porosity [62], including improved proliferation and migration of SCs [63], as well as beneficial effects on macrophages [64,65] and angiogenesis [66]. Further improvements in fabrication processes will allow finer control of the porous structures of conduits and more precise regulation of cellular infiltration, nutrient exchange, and metabolic waste elimination at repair sites [62]. As an example, NCs with selective, unidirectional permeability afforded by asymmetric walls of small internal pores and larger external pores provide a promising approach toward spatially regulating substances delivered within the conduit, such as drugs or growth factors [67]. These build on earlier work showing pore size to be a crucial determining factor in allowing nutrients and other molecules to enter the lumen while limiting the entry of cells such as fibroblasts [35,68]. Other characteristics affected by porosity include mechanical performance, biodegradation rate, and cellular adhesion and proliferation [62].

#### 2.1.4. Natural or Synthetic Composition

NCs can be fashioned from natural or synthetic materials or a combination of both as hybrid composite biomaterials. The characteristics of commonly employed materials in experimental nerve repairs and their relative advantages and disadvantages have been extensively reviewed, including several more recent published works [14,69,70,71,72,73,74,75]. Though critical to those developing and refining NCs, the intricacies of their properties are largely excluded from this review in the effort of providing a clinically focused discussion.

Briefly, naturally sourced materials used in NCs are numerous, and those that are well-studied as components for NCs either alone or in combination with another natural or synthetic polymer include collagen [76], chitosan [77], silk [78], gelatin [79], keratin [80], hyaluronic acid [81], alginate [82], fibrin [83], and agarose [83], among others. Natural materials are generally viewed as enabling less nondenaturing bioreactivity in the body, with generally better biocompatibility and a lower risk of degradation products exhibiting toxicity [84].

Despite the above, NCs of exclusively natural composition have largely been replaced by synthetic or composite approaches, which leverage the manufacturability, strength, and customization allowed by synthetic polymers. Among synthetic options, well-studied biodegradable polymers include polyurethanes such as PCNU (polycarbonate urethane) and PEUU (poly(ester urethane) urea); polyesters such as PGA (poly(glycolic acid)), PLA (poly(lactic acid)), PLGA (poly(lactic acid-co-glycolic acid)), and PCL (polycaprolactone); along with many others. Nonbiodegradable polymers include, among others, silicone, poly(tetrafluoroethylene), and polystyrene [69].

### 2.2. Delivery of Therapeutic Agents

Modern NCs offer many functional applications in their composition, and a particularly translatable method of improving nerve regeneration after repair is the use of biomaterials for the delivery of locally acting therapeutics. The characteristics of the ideal drug delivery conduit are a subject of frequent review [10,41,58,85], and the generally agreed upon characteristics of implantable drug delivery devices for nerve regeneration include (i) biocompatibility, (ii) mechanical characteristics of flexibility, adequate strength, and suturability, (iii) porous, (iv) degradation products with low bioreactivity themselves relative to the target therapeutic’s function, (v) relatively low foreign body responses, and (vi) easily scaled to manufacturing.

The use of biodegradable synthetic polymers for drug delivery is particularly salient in nerve repair, as many promising bioactive substances shown to improve neural regeneration have either unknown or harsh side effects when administered systemically. Drug-loaded polymers are synthesized through many methods, and the increasing accessibility of the staples of NC fabrication techniques such as electrospinning, 3D printing, and bioprinting has opened the field to many research groups previously excluded without dedicated collaborators in biomedical engineering.

#### 2.2.1. Cellular Approaches

As noted previously, SCs play a pivotal role in the healing of peripheral nerves, particularly when regeneration is required across a large nerve gap (>3 cm), such as those created by nerve transection and the subsequent retraction of nerve ends. Cultured SCs have been delivered to transected peripheral nerves in in vivo models since as early as 1979 [86,87], and the fabrication techniques of modern conduits aim to more closely mimic the extracellular environment to best allow SC proliferation and migration.

Direct delivery of SCs has shown success in many in vivo models, particularly in recent years, often through the injection of matrices seeded with cultured SCs [26,88,89], or through the inoculation of the walls for porous conduits [25]. These approaches have shown improved performance in metrics such as axonal regeneration, elongation, and myelination, occasionally approaching regeneration similar to autografts. Further illustrating this point, a recent systematic review by Vallejo et al. found that PNI repairs involving SC-loaded nerve guides over a gap of at least 10 mm showed similar results to autograft controls in histomorphometric and functional outcomes [90].

Unfortunately, inherent limitations exist in the use of primary SCs for delivery, particularly in PNI repair where acute trauma accounts for the vast majority of cases [91]. Transplantable primary SCs must be donor derived to avoid graft-versus-host interactions, and culturing cells to the quantity needed for human applications is time-consuming and often not feasible for adoption into widespread clinical use by current methods [92]. Though providing a supportive environment for the growth of endogenous SCs remains of vital interest, alternative cellular options have been considered for the delivery of exogenous cells, including SC precursors and stem cells of many types.

Stem-cell-based therapies represent a promising approach to the temporal limitations of culturing primary SCs. Ethical and safety concerns outside of the scope of this review are associated with the use of human embryonic stem cells given their source, despite their potential utility in the generation of SCs via rapid proliferation [93]. Induced pluripotent stem cells do not pose the same ethical issues and have shown promising results in generating large numbers of functional SCs through a SC precursor intermediate [94]. However, there are concerns with respect to their tumorigenic potential. Other prominent stem cell types under investigation for their role in traumatic peripheral nerve injury have been recently reviewed by Kubiak et al., with numerous in vivo studies showing potential for clinical application [95]. Oftentimes, these cells exert their effects by exhibiting a Schwann-cell-like phenotype or through the secretion of growth factors, and examples of frequently employed cell lines include adipose-derived stem cells (ADSCs) [83,96], bone-derived mesenchymal stem cells (BM-MSCs) [97], and neural stem cells [98], though long-term safety and efficacy studies are still needed to compare the wide range of considerations to determine which is superior. Several studies have compared ADSCs and BM-MSCs, with evidence suggesting that ADSCs may provide slight advantages in addition to greater ease of harvest, wide differentiation potential, and low immunogenicity [99,100]

#### 2.2.2. Neurotrophic Agents/Growth Factors

An alternative to implanting entire cultured cells at a nerve repair site is to instead leverage only the growth factors they secrete. Numerous neurotrophic agents and their impacts on both in vitro and in vivo models have shown beneficial effects associated with their use, though none have been integrated into clinical practice for the treatment of peripheral nerve repair to date. Several of these factors are found at low concentrations physiologically; however, they are upregulated in neurons and denervated SCs upon nerve insults, leading to the frequent study of how their benefits can be applied or enhanced at sites of nerve repair [101]. Unfortunately, as proteins rather than shelf-stable pharmacologic agents, a particularly challenging aspect of applying neurotrophic factors as an adjunct therapy in nerve repair is their instability. Historical issues with their use include rapid inactivation by enzymes leading to short half-lives, as well as reaching adequate concentrations at their intended sites without triggering dose-limiting side effects [102].

Advancements in culturing methods and the design of biocompatible scaffolds have allowed for the incorporation of growth factors into numerous biomaterial-based devices with promising results. As recently reviewed by Wan et al., nerve growth factor (NGF), brain-derived neurotrophic factor (BDNF), neurotrophin-3 (NT-3), glial-derived neurotrophic factor (GDNF), insulin-like growth factor-1 (IGF-1), basic fibroblast growth factor (bFGF), and vascular endothelial growth factor (VEGF) remain among the most widely examined factors for this purpose, having been employed in varieties of NCs, hydrogels, microspheres, nanoparticles, and exosomes [103].

Within the realm of peripheral nerve pathology, recombinant human nerve growth factor (rhNGF) has undergone evaluation at different phases of clinical trials as a treatment for diabetic neuropathy (phase III) [104], HIV-associated sensory neuropathy (phase II) [105], and most recently, neurotrophic keratopathy (NK) (phase IV) [106]. However, only topical rhNGF for NK has materialized as an FDA-approved therapy. Notably, the subcutaneous administration of rhNGF for diabetic neuropathy, to a large sample of 1019 patients, failed to show a significant benefit and resulted in unexpectedly high rates of injection site pain, hyperalgesia, myalgia, and peripheral edema when compared to that of the expected outcome from phase II testing [104]. As a result, over the past decade, the majority of research into the applications of biologic therapies has centered on local delivery, directly at the site of repair.

From the application of its existing indication for the treatment of NK, NGF may face fewer hurdles in regulatory agency approval and translation to nerve repair relative to other growth factors. Further study is necessary to elicit any adverse effects of local delivery, particularly any effects similar to those seen in the systemic administration for humans, relative to nerves. These include pain, myalgia, and edema, which may be difficult to appreciate in animal models without deliberate observation, or impossible altogether prior to human use.

#### 2.2.3. Pharmacologic Agents

The delivery of existing pharmacologic agents to aid in nerve healing after repair offers several distinct advantages over the delivery of other substances such as neurotrophic factors and cells. In terms of logistics and the ease of translation into clinical practice, the use of well-studied drugs with years of successful application in other indicated uses should allow for faster translation into clinical practice, while simultaneously hosting readily available sources for manufacturing. Among the many drugs studied to aid peripheral nerve regeneration, immunosuppressants, corticosteroids, and drugs with antioxidant or anti-inflammatory effects have frequently been delivered in nerve conduits, scaffolds, and hydrogels.

Drugs can be better suited to endure the often harsh fabrication processes of conduits without losing function. In contrast, the fabrication of NCs containing biological agents such as proteins and cells often requires additional considerations in manufacturing, storage, and application to prevent their exposure to potentially damaging conditions such as high temperatures, organic solvents, adverse reactions with biomaterials or their degradation byproducts, and other conditions that deviate from near-physiologic conditions [56,102,107]. Preclinical outcomes from the successful embedding of cells and growth factors are representative of the great strides taken in regenerative medicine in recent decades. However, the introduction of established pharmaceuticals with well-known side effect profiles and interactions in the body into NCs likely remains closer to translation into surgical practice.

An example of the qualities described here as well as additional benefits provided by drug-specific delivery in nerve repair lies in the immunosuppressant tacrolimus (FK506). This drug is particularly relevant to peripheral nerve repair given its historical use in combatting the immunogenicity of nerve allografts prior to the introduction of acellular allografts, as discussed later in this review [108]. Numerous groups have examined this FDA-approved calcineurin inhibitor for its benefits in nerve healing since their description in 1994 [109], with several recent publications describing its application in a drug-releasing nerve wrap in in vivo models [34,110,111]. Local delivery of this drug has been shown to increase the number and size of myelinated nerve fibers, increase the number of regenerating motor and sensory neurons, and improve functional recovery, in addition to providing local immunosuppression at repair sites [34,112]. Tacrolimus is frequently prescribed for this purpose in organ transplant recipients, though its harsh side effects—including tremors, headache, nephrotoxicity, and diabetes mellitus—preclude its regular systemic administration as a surgical adjunct for peripheral nerve repair [113,114]. When incorporated into NCs, tacrolimus maintains its bioactive properties through the high voltages and often harsh solvents involved in electrospinning, and the drug–polymer matrix produced through this process is thermally stable at ambient and body temperatures, facilitating both storage and clinical application, respectively [34]. As such, tacrolimus-releasing conduits and wraps have emerged as promising and readily translatable therapies in peripheral nerve surgery.

The delivery of pharmacologic agents to injured nerves is yet another topic that is frequently reviewed in the literature [71,115,116]. However, comprehensive reviews of this subject often include pharmacologic agents that are not FDA-approved. Additionally, the heterogeneity seen in preclinical models of nerve injury adds further complexity to performing comparative studies, even when the same animal model and nerve are used. As such, we have assembled Table 1 below to include only drugs that (i) hold current FDA approval for any indication, (ii) have been delivered locally by a nerve conduit, hydrogel, or scaffold, and (iii) have been applied in a full transection and repair in vivo model. In doing so, we hope to highlight approaches that may be the closest to clinical adoption: those involving established drugs, for use in repairs across full transections, which frequently employ conduits.

In addition to passive drug delivery based on the degradation of polymers, advances in biomaterials have allowed for systems with finer control, with one example being the use of magnetic nanoparticles (MNPs). These functions are achieved by either embedding MNPs within hydrogel networks or electrospinning fibers of other, more biocompatible polymers infused with MNPs [127]. Potential applications to the peripheral nervous system include neural cell manipulation and guidance, as well as the spatially precise delivery of bioactive compounds [127]. For instance, Giannaccini et al. were able to conjugate NGF and VEGF to MNPs prior to injection into a nerve conduit with a strip of magnetic tape around their center. These growth factors have short half-lives which limit their use, and their distributions are nonuniform within regenerating nerves due to being secreted from the proximal and distal nerve stumps. Through the use of NMPs for delivery, these growth factors were maintained at a higher concentration within the conduit, extending their use, and were particularly concentrated in the region of the nerve deficit most in need, the center [128]. Though promising in theory, the distribution of magnetic metals in a high surface area to volume nanoparticle form within the body will require significant additional study and proof of safety before translation into human applications. MNPs pose significant risks to multiple body systems upon exposure, and well-established interactions include the activation of oxidative stress, inflammation, and indirect DNA damage [129]. As such, their use remains a contested topic within the discussion of drug delivery, as the benefits of sequestering the delivered drug must outweigh the inherent inflammatory reactivity of introducing such metallic materials into the body.

In addition to precise spatial delivery, recent advancements in drug encapsulation within biomaterials allow greater temporal control, such as through ultrasound-responsive delivery systems employing multiple bioactive compounds. This was recently demonstrated by Shan et al. by encapsulating a hydrogel network directly with a drug shown to mitigate neuroinflammation in the early stages of repair (vitamin B12), as well as with microspheres loaded with nanoparticles, which are loaded with factors capable of promoting long-term regeneration (NGF) [130]. Drug release is then regulated by ultrasonic stimulation, for the regulation of both rapid release from the loose hydrogel structure and prolonged release from the more dense triple-encapsulated system according to the ideal therapeutic time window [130].

## 3. Intraoperative Electrical Stimulation

### 3.1. Current Approach

The electrical stimulation (ES) of nerves to promote axonal regeneration is a relatively recent addition to the study of nerve repair and healing, with few clinical trials published thus far [131]. The application of electrical currents to nerves has been leveraged by many medical fields, with recent and exciting results in interventions ranging from modulating pain [132] to improving memory [133]. Within the realm of nerve repair, the foundational work of using ES to accelerate axonal regeneration is often attributed to studies by Hoffman [134], Nix and Hopf [135], and Pockett and Gavin [136].

The most salient application of ES to surgical nerve repair lies in the stimulation of nerves through direct contact with electrodes to generate its beneficial effects. However, as this method often uses the relatively non-specific phrase of “electrical stimulation” in its publications, a discussion of other, similarly named therapies is warranted. In humans, therapeutic electrical stimulation has been applied to patients with peripheral nerve pathology in four particularly similar methods: (i) at low intensity through the skin, largely to treat pain and neuropathy (transcutaneous electrical nerve stimulation, TENS) [137]; (ii) also through the skin, however directed at denervated muscles following peripheral nerve injury, largely to mitigate atrophy and preserve function upon reinnervation (electrical stimulation of denervated muscle, ESDM) [138]; (iii) stimulation through implants in downstream denervated musculature after nerve injury for the same purpose as the previous method [139]; and (iv) directly to injured nerves to promote reinnervation of downstream targets, as discussed in the remainder of this review (commonly referred to simply as electrical stimulation, ES). Of note, nomenclature regarding approaches (ii), (iii), and (iv) varies frequently in reporting and would benefit greatly from standardization within the future literature. The use of 20 Hz neuronal electrical stimulation is suggested.

ES in the context of this discussion is applied proximal to a repaired nerve site, either intraoperatively before skin closure [140,141,142] or in the immediate postoperative period [143]. It is performed most commonly at a frequency of 20 Hz for one hour, a frequency and duration first shown effective by Al-Majed et al., which has since been confirmed by numerous animal [144,145,146] and human [140,141,142,143] studies. This significantly increases the levels of cyclic adenosine monophosphate (cAMP) within neurons [147,148,149], upregulates neurotrophic factors and their receptors on both SCs [150,151,152] and neurons [153], and increases the levels of growth-associated genes and cytoskeletal proteins necessary for growth [145,154,155]. The specific directionality of the stimulus is applied to trigger a cascade of action potentials toward the cell body (antidromic), as this has been shown to be the site of action of ES’s benefits [156].

The first randomized control trials of one-time, intraoperative ES took place in 2010 by Gordon et al. and examined intraoperative ES as a method to improve outcomes in patients with median nerve compression secondary to carpal tunnel syndrome (CTS) [142]. With the application of 20 Hz of stimulation for one hour during carpal tunnel release, motor and sensory reinnervation significantly improved, though a benefit to functional recovery was not found. The next major trial of ES involved patients undergoing surgical repair of complete transection injuries to digital nerves and found that the same ES parameters significantly improved sensory outcomes in all methods tested, though did not show a significant disability improvement as assessed by the Disability of the Arm, Shoulder, and Hand questionnaire (DASH) compared to the control, sham ES [141].

Further study has led to intraoperative ES being applied to oncologic neck dissection to mitigate postoperative shoulder dysfunction [140], as well as decompression operations to treat cubital tunnel syndrome [143,157]. These studies have further supported the efficacy of ES shown in prior work, in addition to providing evidence of functional benefits to the disability and recovery of muscle strength.

To our knowledge, only one human study, by Wong et al., discussed above, has involved the use of ES in the treatment of a complete transection and repair of a peripheral nerve [141], with the remainder involving comparatively minor injuries from compression [142,143,157] or axonal injury due to devascularization and retraction [140]. Less severe nerve insults such as those that are close in proximity to their targets of innervation are less likely to experience the poor outcomes ES has been developed to address, and as a result, its full effects may be revealed only in severe injuries with historically worse healing. Though Wong et al. applied ES in a population meeting these characteristics [141], transections of nerves in the compact space of the digits often also involve concomitant tendon, artery, or vein injuries, and this heterogeneity of patients may have reduced the study’s power to detect a significant difference. As such, further study in more severe injuries which are further from their reinnervation targets, and with more homogenous patient populations, will be valuable to establish the true efficacy of ES in improving peripheral nerve repair outcomes.

### 3.2. Novel Biomaterial Applications to Electrical Stimulation

As the effects of electrical stimulation on axonal regeneration have been shown in animal studies since as early as the 1950s [134], many groups have developed biomaterial-based approaches for its delivery, with many of these progressing to in vivo studies with success. However, very few, if any, have successfully made it to human use, as electrically conductive polymers for highly reactive materials are relatively nondegradable, and thereby pose long-term foreign-body response issues for patients at this time. Among others, these include conductive, self-powered, and wireless electrically stimulating nerve conduits and devices.

#### 3.2.1. Conductive Polymers

Numerous electrically conductive NCs have been studied in animal models, with particularly promising results when combined with ES, though none have reached human implementation at this time due to concerns largely related to biocompatibility [158]. Materials used for this purpose include polypyrrole (PPy) [27,159,160,161], polyaniline (PANI) [162,163], and poly(3,4-ethylenedioxythiophene) (PEDOT) [164], though these materials are largely limited to integration within another biocompatible natural or synthetic polymer at this time. Carbon nanotubes (CNTs) and graphene (GO) represent additional materials being studied for their applications as conductive additives for enhancing neural growth and regeneration, though their safety profiles with respect to in vivo application are less clear at this time [158]. Notably, graphene-based materials have also been studied for their ability to induce the transdifferentiation of MSCs to SC-like phenotypes through the delivery of electrical stimulation alone, in the absence of chemical growth factors [165].

#### 3.2.2. Self-Powered Conduits

Similar to conductive NCs in intended effect, self-powered conduits incorporating nanogenerators, which are able to generate electric potentials on their surface upon mechanical deformation, represent a potential source of therapeutic electrical stimulation that would not require an external electric source. Instead, the conduits may generate an electric stimulus through activation by forces such as natural body movement, noninvasive ultrasound waves, or magnetic fields, as recently reviewed by several groups [166,167]. Other examples of the potential applications of self-powered conduits include the incorporation of piezoelectric polymers to generate electrical power from mechanical deformation caused by natural rat body movement [168] or from physiologic actions such as breathing [169].

Promising materials under study for these applications are numerous [166], with examples including piezoceramics such as zinc oxide (ZnO) [168], as well as synthetic piezopolymers such as the nonbiodegradable polyvinylidene fluoride (PVDF) [170] and biodegradable (PHBV) [171] and poly(L-lactic acid) (PLLA) [172]. Similar to concerns discussed previously in regard to magnetic nanoparticles, the use of some conductive and piezoelectric materials in nanoparticle form will require extensive study for proof of safety before implementation in humans. As an example, ZnO nanoparticles, even at low concentrations, can induce oxidative stress and damage DNA in cells [173]. This may be mitigated by implementing the gradual release of such particles when incorporated into slowly degrading polymer substrates [168], though highlights the risks involved in the introduction of functional exogenous substances into the body, even locally.

Some natural biopolymers host relatively strong piezoelectricity including cellulose, chitosan, and collagen [174], and continued study of these compounds in this context is likely warranted.

#### 3.2.3. Wireless Nerve Stimulators

A limitation of the most commonly employed ES protocol (20 Hz, 1 h) at this time is the need to extend a completed operation by a full hour in order to allow stimulation by direct contact with nerves. This increases operating times, as well as risks such as exposing the patient to greater amounts of anesthetic and the potential for complications. As one method to combat this, biocompatible electrode advancements have allowed the development of wireless, resorbable nerve stimulators that can be implanted at a nerve repair site immediately prior to wound closure and activated when most appropriate. These stimulators are able to stably deliver therapeutic levels of ES (20 Hz for 1 h) prior to the rapid degradation of its electronic components within 3 days and total degradation of the device within 10 days [175]. As the optimal ES frequency and duration are still debated, other groups have designed wireless nerve stimulators that are able to function for up to 6 days prior to resorption within 25 days [176], as well as up to 30 days with degradation within 50 days [177].

At this time, the utility of ES outside of the immediate postsurgical period is still debated; however, if greater efficacy is shown by extended or continuous ES, these implantable devices will likely be crucial to effective clinical translation.

## 4. Nerve Grafts

### 4.1. Currently Available Nerve Grafts

In cases of severe nerve deficits where joining the distal and proximal stumps cannot be accomplished without tension, the use of a grafted nerve segment interposed between the transected nerve stumps may be necessary. Clinically, current options include the gold-standard autologous nerve autograft or commercially available processed acellular allograft.

#### 4.1.1. Autograft

The first described use of successful autografting in an animal model was by Philipeaux and Vulpian in 1870, using an autologous lingual nerve graft to repair a hypoglossal nerve deficit in a dog [178] (per [179]). The first successful human autograft is less clear, though may have been by Robson in 1889, who bridged a median nerve gap with a popliteal nerve graft [180,181]. Repair by autograft remains the current gold standard due to the graft’s innate architecture and resident SCs releasing growth factors providing the adequate environment necessary to facilitate nerve regeneration [182,183].

Though replacing a deficit with a freshly harvested, autologous nerve has a proven history of success, this carries distinct disadvantages in application. Along with the inherent morbidity of removing a functional nerve used as a donor, this second surgical site presents additional opportunities for infection, can require intraoperative repositioning to access, prolongs time under anesthesia, and increases the potential for painful neuroma formation at the donor site [184]. According to a recent review evaluating 214 sural nerve harvests for grafts, 92.5% of these patients experienced sensory deficits, 22.9% chronic pain, 1.4% wound infections, and 7.0% wound complications other than infection [185]. Additionally, necrosis of resident SCs as a result of decreased perfusion to the graft can take place after transfer, which may be problematic in particularly long repairs [186].

Nerve autografts can be size-matched to the recipient nerve diameter by choice of donor nerve, as well as by using single, cable, trunk, or interfascicular techniques [182,187]. The donor nerves most commonly used for this include the sural nerves, medial and lateral antebrachial cutaneous nerves, superficial branches of the radial nerve, dorsal cutaneous branches of the ulnar nerve, superficial and deep peroneal nerves, intercostal nerves, and the posterior and lateral femoral cutaneous nerves [185,188].

A recent systematic review comparing available, FDA-approved NCs and wraps against direct repair or autograft in upper limb peripheral nerve repairs found that the use of such devices may be associated with a higher rate of adverse events and greater need for revision surgery, though the evidence was noted to be very uncertain [189]. There was little or no difference in mean sensory recovery or integrated functional outcome scores at 2 years between the groups, but once again, both conclusions were stated to be by very uncertain evidence. Though five-year functional outcomes may be slightly improved with device use, this was additionally with low-certainty evidence [189].

#### 4.1.2. Allograft

In addition to their animal experimentation in autografting, Philipeaux and Vulpian also conducted early, though unsuccessful, work in allografts in 1863 [179]. The first attempted use of an allograft in humans was likely by Albert in 1878, using nerve material from a recently amputated leg of another man to bridge a gap in the median nerve of a patient, which became necrotic and required removal [190] (per [181]).

Allografts only became a viable clinical option with the development and further understanding of therapeutic immunosuppression, though the long-term systemic immunosuppression needed to prevent rejection of a cadaveric nerve caused concern for other pathologies, including organ toxicity, opportunistic infections, and neoplastic processes [191]. Fresh allograft use has typically been described only in severe cases where the extent of nerve deficit would be otherwise irreparable by available donors. This is illustrated by the bridging of a 23 cm sciatic nerve defect using 10 cabled fresh allografts followed by immunosuppression in an 8-year-old boy by Mackinnon et al. in 1988 [192]. Similar repairs were performed in seven additional patients who required immunosuppression for an average of 18 months [108]. This was circumvented by the introduction of processed acellular nerve allografts (ANAs) to clinical practice in 2007, which do not require systemic immunosuppression, and had spread in use to approximately 70% of hand surgery practices in the United States by 2018 [193].

Processed, in this setting, describes the removal of immunogenic components from an initially cellular nerve graft, leaving behind a highly organized extracellular matrix capable of serving as a scaffold that closely mimics the native tissue architecture [194]. Nerve guides containing acellularized epineurium, fascicles, and endoneurium to support growth, migration, and angiogenesis provide a platform for nerve regeneration [184,195]. Though necessary in order to fully combat immunogenicity, this process also removes SCs from the graft, resulting in the ANA still ultimately being reliant upon in situ SC migration similar to other non-SC-loaded nerve conduits [196,197]. Though ANAs are architecturally primed for this migration, the lack of adequate functional SCs may be one of many limiting factors for their application to longer nerve deficits than their current maximum factory-available length of 7 cm [188,196].

Another established limitation of ANAs in bridging nerve gaps of greater lengths is the natural senescence SCs undergo when exposed to environmental stress, aging, or chronic denervation [198,199,200]. Strategies to mitigate the senescence of in situ SCs within ANAs as they proliferate and migrate long distances to reinnervation targets are currently under study, including the neurotrophic effects of side-to-side bridge grafting [201] and reverse end-to-side (RETS) nerve transfers [202,203], which have shown benefits in muscle recovery. Both tacrolimus and electrostimulation, discussed previously, have also been suggested to promote earlier SC reinnervation in the distal stump of transected nerves [199]. Finally, the use of TGFβ, interleukin-10 (IL10), and other cytokines mediating the interactions between SCs and macrophages recruited to the repair site are under study for their abilities to help maintain the repair-oriented SC phenotype, particularly in delayed PNI repair [199,204].

### 4.2. Advancements in Grafts

#### 4.2.1. Xenografts

Xenografting refers to the process of transferring donor tissue of one species into a recipient of another species. Similar to allografts, improved tissue-processing and immunosuppression techniques in recent decades have allowed xenografts to emerge as a potential substitute for autografts in peripheral nerve repair. In addition to eliminating concerns of donor site morbidity and the need for immunosuppression previously solved by allografts, xenografts host the advantages of being easily scaled to manufacturing, abundant in sourcing, and cost-effective in comparison to ANAs which require cadaveric donor tissue [205].

Xenograft donor tissue is often from bovine or porcine sources, with the latter often favored due to widespread availability, relative genetic similarity to humans, and a thorough understanding of porcine physiology through decades of biomedical study as a model organism. In addition to this, they are largely anatomically compatible with humans, sharing a similar nervous system structure [33]. Further evidence for the utility of porcine tissue as it applies to nerve healing lies in the currently FDA-approved nerve conduit Surgisis nerve cuff, which is composed of a cell-free collagen matrix sourced from porcine small intestinal submucosa [206].

Several studies have been performed in animal models over the past decade showing promising results, including the repair of a 6 mm facial nerve deficit in rats using an acellularized rabbit xenograft, which showed comparable results to autografts [207]. In another similarly designed study bridging a 1 cm deficit, it was found that autografts outperformed xenografts and allografts, with these two options performing similarly [32]. Recently, experiments have shown autografts to be superior to allografts, which in turn are superior to xenografts when applied to a 15 mm deficit. Xenografting was achieved in this latter study by utilizing processed human donor tissue in a rat nerve deficit [208].

Decellularization techniques are still being modified, showing improved results that may eventually match or overtake results from the gold-standard autologous graft. One such example includes the use of a novel supercritical carbon dioxide extraction technique, which was applied in tandem with the well-established chemical decellularization seen in commercial acellular nerve allografts to generate porcine xenografts for implantation into rats. The xenograft processed in this manner was compared against a hollow nerve guidance conduit, a chemically decellularized only porcine xenograft, and an autologous graft, showing a recovery similar to an autograft and significantly better than the nerve conduit or chemically treated only xenograft in a 15 mm gap [33].

#### 4.2.2. Fat Grafting

As mentioned previously, adipose-derived stem cells are a frequently employed SC-like cell currently being investigated for delivery in biomaterial conduits and hydrogels as an alternative to SCs, showing benefits comparable to an autograft in several metrics [83,96,99,100]. In addition to this use, autologous fat grafting has also been explored in nerve repair, employed in a manner similar to that seen in several cosmetic procedures; autologous fat is harvested, minced, centrifuged, and injected around a nerve repair. This has been shown to improve remyelination, fiber density, and axon count, as well as provide benefits to pinprick sensation, motor sensory recovery, SC migration, and inflammation [209,210,211]. These effects have been attributed to neurotrophic secretions of ADSCs, as well as potential proangiogenic and immunomodulatory signals that may help to decrease neuropathic pain and suppress neuroinflammation [212]. Though not a “graft” in the sense of serving as a connection between nerve stumps, this likely represents another approach that may be applied synergistically for clinical benefits in nerve repair.

## 5. Commercial Availability

At this time, several nerve conduits have been cleared by the FDA for use in peripheral nerve repair, with most being allowed through a 510 (k) pathway by proving substantial equivalency to an existing device [213]. A summary of the characteristics of these devices can be found in Table 2. Of these, the vast majority are simple, hollow tubular designs or wraps to encircle a repair site, though recently, options incorporating additional internal structures have been introduced. These include Nerbridge (Toyobo Co. Ltd., Osaka, Japan), which hosts an inner collagen matrix, and Neuragen 3D (Integra LifeSciences Co., Princeton, NJ, USA), with a collagen/chondroitin-6-sulfate inner hydrogel matrix [214]. Another structural characteristic of a recently approved wrap (NerveTape, BioCircuit Technologies, Atlanta, GA, USA) is the use of microhooks of nitinol, which allow for repair without the use of sutures to secure the nerve ends or attach the wrap to the repaired nerve, per the manufacturer’s website.

The majority of available NCs are composed of porcine- or bovine-derived collagen, often from the small intestinal submucosa, though pericardium and skin have also served as collagen sources. Other natural materials employed in commercially available NCs include calcium alginate/hyaluronic acid and chitosan. Among synthetic materials, PVA, PLCL, and PGA conduits have been approved for use [213,214,215].

Only one acellular nerve allograft is commercially available for use in peripheral nerve repair at this time, the Avance Nerve Graft (Axogen, Inc., Alachua, FL, USA), which is composed of decellularized human cadaveric nerve.

## 6. Comparative Studies

Given the heterogeneity involved in injury characteristics, surgical techniques, and institutional approaches to nerve repair, high-quality comparative studies are difficult to carry out and frequently reach inconsistent conclusions. For instance, a recent systematic review and meta-analysis of in vivo preclinical models found autografts to be superior to ANAs in seven of eight assessed outcomes [216]. In humans, a recent systematic review reported being unable to draw any conclusions regarding comparisons of ANAs, NCs, and autografts due to very low certainty evidence [217]. A systematic review by Lans et al. [218] evaluating currently employed NCs, ANAs, and autografts found that ANAs and autografts were superior in restoring meaningful recovery compared to NCs, though not significantly different from one another. The authors also performed a cost analysis, finding that the total cost of allograft usage was less than autografts in inpatient settings, but comparable in outpatient settings [218]. Their results added further support to conclusions detailed by Mauch et al. in a systematic review performed 3 years prior [219]. Though limited to digital nerve repair exclusively, evaluating the three methods in addition to primary repair showed autografts and allografts to perform similarly in return of sensation, whereas NC repairs more often resulted in poor sensory recovery, as well as a higher rate of complications [219]. Taken together, these findings could support the further adoption of ANA use not because of a clear benefit but rather noninferiority, particularly when consideration is given to the increased cost, prolonged operating time, incapacitation of a donor nerve, and increased risk for neuroma formation necessitated by autologous grafting.

## 7. Further Study and Synergistic Applications

Though ANAs have shown promise in comparison to autografts and NCs thus far, several unforeseen circumstances have arisen that will require further exploration and potential refinement. These include cases in which a definitive cause of allograft failure cannot be found, poor recovery in repairs of longer lengths or larger diameters, and other generally abnormal performances of ANAs [220,221]. Examples of these abnormal performances include isolated reports of the reabsorption of ANAs postoperatively without noticeable recovery [222], a report of “regenerated cable” formation with only minimal axonal regeneration that improved upon revision and autograft [223], and abnormal neuroma formation [221]. As such, continual outcome reporting and analysis will be vital to improve methods of ANA application or refine more specific indications for their use if necessary.

Future applications of biomaterials appear promising provided the benefits seen in early in vitro and in vivo studies hold true as they are translated to clinical practice. Relatively few NCs have progressed to clinical trials [224] relative to the immense amount of NCs that have been proposed and developed. A major contributor to this especially high hurdle is the fact that for the majority of the time the applications mentioned in this review have existed, autografts have remained superior in almost every metric, aside from morbidity associated with a donor site. However, with conduits, grafts, and other surgical adjuncts nearing the efficacy of the gold-standard autograft in some in vivo models, it is likely that the near-century study of complex neural repair mechanisms, material properties, and tissue processing may soon yield repairs rivaling and potentially surpassing those of autografts. This will be a welcome shift away from an intervention that causes some degree of patient harm by design and delivers relatively poor postoperative outcomes relative to many interventions in the wider fields of surgery and medicine.

It is not difficult to imagine a future nerve repair that leverages aspects of each intervention mentioned here, such as that illustrated in Figure 2. Though most have been developed as controlled, separate variables, this does not preclude their use synergistically except in select cases. For example, a hypothetical nerve deficit of critical length could be bridged with an affordable and safe acellular xenograft, with ends flanked by bioprinted fascicles meeting the exact specifications needed for a near perfectly aligned repair. This graft could be pre-seeded with Schwann or Schwann-like cells with the ability to replicate, migrate, and myelinate with temporal precision through the controlled delivery of drugs and/or growth factors, and be encased by a wrap of conductive or piezoelectric polymers. In addition to this, electrical stimulation could be applied intraoperatively or in the immediate postoperative period by a biodegradable, implantable nerve stimulator. Each of these interventions have shown promise in respective in vivo models, with many nearing human implementation.

## 8. Conclusions

Efforts to improve nerve healing are multidisciplinary, drawing input from medical experts, biomedical engineers, and material chemists, and novel approaches leverage a wide variety of advancements in countless fields and specialties. Many interventions to supplement traditional end-to-end repairs have been shown to improve in vivo healing with immediate clinical relevance. These have been made possible by advancements in the safety of biomaterials, the properties instilled upon them in their fabrication, and the functionalization of many natural and synthetic polymers. Among these, the local delivery of bioactive agents that would otherwise produce harsh or unknown side effects has created new opportunities for improving the complex pathways of neural regeneration. Additionally, the electrical stimulation of nerves in the perioperative period has been shown to be beneficial in clinical trials, and the incorporation of safe biomaterials to supplement its delivery may further increase its efficacy and translation into clinical practice. Finally, acellular nerve allografts have allowed for results approaching those of autografts, and with further refinement, may allow for the widespread and cost-effective adoption of an intervention that eliminates the morbidity of harvesting a donor nerve.

Returning to the paper quoted in the introduction of this review, the remainder of the paragraph published in 1944 deserves consideration in this light:

“*An unbiased survey of existing methods of nerve repair… shows plainly that no one of them is sufficiently superior to the others to deserve a monopoly of attention. In times of urgency such as these, the weighing of one method against another had therefore better give way to a concerted effort to extract the best features from all available methods and combine them to the best practical advantage*”[1]

No one intervention discussed in this review has eliminated the need for improvement allowed by the others. In time, the synergy allowed by their combination may revolutionize the treatment of peripheral nerve injuries, and ideally, be supplanted by future, unforeseen innovations.

## Figures and Tables

**Figure 1 bioengineering-11-00776-f001:**
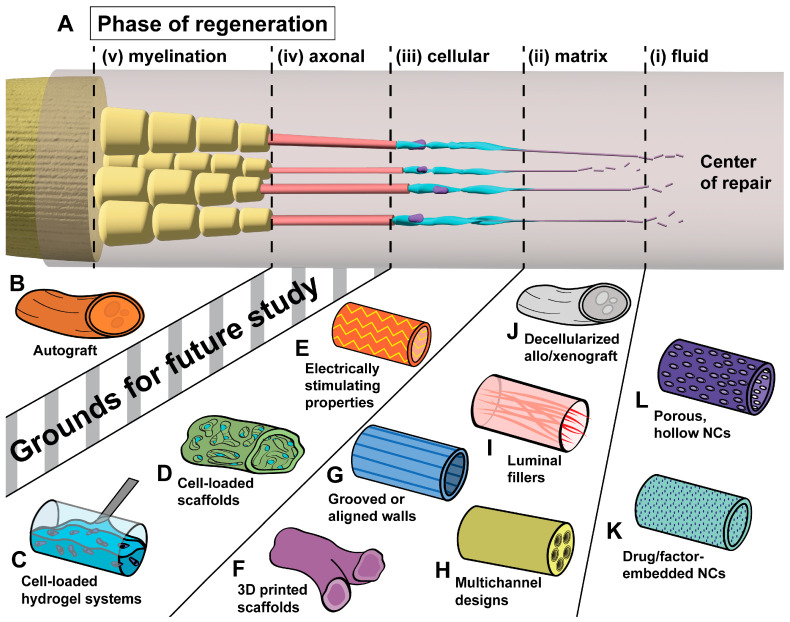
The five phases of neural regeneration across a gap and selected implants to enhance regeneration in each. (**A**) Healing across a nerve deficit begins with (i) the secretion of ECM precursors, which (ii) coalesce to a matrix of ECM proteins, allowing (iii) cellular traversal of the deficit. (iv) These cells guide axonal growth, which is followed by (v) myelination [8,9,10,11]. (**B**) The current gold standard of repair across a deficit, autologous nerve graft, obtained through harvest of a patient sensory donor nerve. (**C,D**) Implantable scaffolds [25] and hydrogel systems [26] may be preloaded with cultured SC or SC-like cells [27]. (**E**) The incorporation of functional polymers that enhance electrical stimulation and conductivity allows for the preservation of denervated SC populations [27]. (**F**) Precise 3D printing of biocompatible scaffolds allows for the design of bifurcating and irregular scaffolds for improved topographic guidance [28]. (**G**–**I**) Topographic guidance features within walls [29] or intraluminal channels [30] and filaments [31] of a conduit encourage migration along its length. (**J**) Decellularization of allografts and xenografts allows for the removal of immunogenic components, leaving behind an ECM scaffold [32,33]. (**K**) The embedding of bioactive substances within conduit walls allows for the controlled, local release of the substance within the repair site [34]. (**L**) Porous, hollow conduits permit nutrient exchange while preventing cellular invasion [35].

**Figure 2 bioengineering-11-00776-f002:**
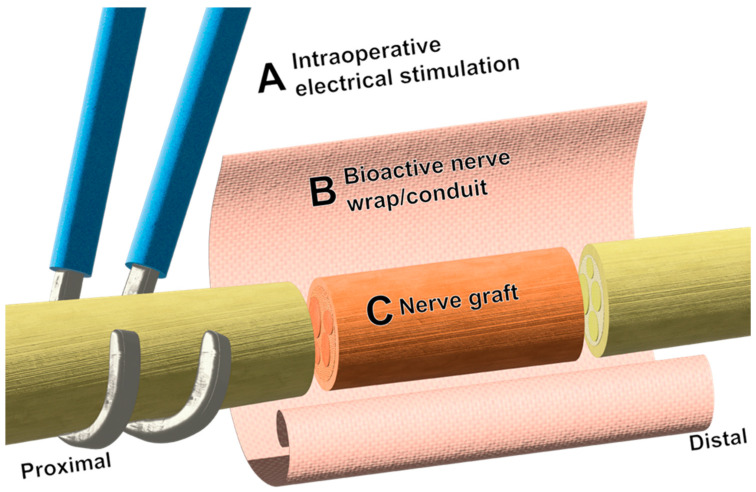
Schematic representation of potential synergistic approaches to nerve repair. (**A**) Intraoperative electrical stimulation is applied proximal to the nerve repair site, commonly by hook electrodes. (**B**) Bioactive wraps and conduits can be fashioned around joined nerve stumps for the delivery of therapeutics such as drugs and neurotrophic factors and have shown promise in preclinical models. Wraps and conduits can also provide protection and topographic guidance cues to regenerating axons. (**C**) At this time, the use of acellular nerve allografts serves as a suitable method to avoid the donor morbidity associated with autografts.

**Table 1 bioengineering-11-00776-t001:** Several pharmaceutical compounds currently approved for conditions unrelated to nerve repair have been delivered locally to in vivo transection and repair models to evaluate their effects on nerve healing across a deficit.

Drug	Drug Type	Material/Fabrication	Model Tested	Study Results	Proposed Mechanism of Benefit
Tacrolimus (FK506) [34]	Immunosuppressant	FK506-loaded PCNU nerve wrap	Rat common peroneal and median nerve transection and repair	Increased number, size, and myelination state of regenerating axons, and increased number of sensory and motoneurons Accelerated recovery of active finger flexion Reduced postoperative nerve swelling	Interaction with Hsp90 through FKBP-52 Guides growth cones of regenerating neurites, accelerating regeneration
Sirolimus (rapamycin) [117,118]	Immunosuppressant	Coated PLGA microspheres in chitosan/collagen nerve scaffold	15 mm sciatic nerve gap	Reduced inflammatory response in bridged area Improved sciatic function index and electrophysiological tests to a level similar to autograft	Interaction with mTOR through FKBP-12 Promotion of autophagy Reduction in scar formation
Cyclosporine [119]	Immunosuppressant	CsA solution loaded in chitosan conduit	10 mm sciatic nerve gap	Improved recovery of muscle mass and function Morphometric evidence of improved axonal regeneration	Interaction with Hsp90 through CyP40 Preservation of myelin
Methylprednisolone [120]	Corticosteroid	MP-loaded microspheres in sustained-release membrane	Rat sciatic nerve transection and repair	Improved myelin sheath thickness, collagen content, and nerve fiber number	Reduces inflammation and scarring at repair site
Dexamethasone [121]	Corticosteroid	Dex-filled silicone nerve conduit	10 mm rat sciatic nerve gap	Improved functional recovery Improved early morphometric results at week 4, though not at weeks 8 or 12	Protects against demyelination and enhances remyelination
Gabapentin [122]	Anticonvulsant	Gabapentin-loaded cellulose acetate/gelatin scaffold	10 mm rat sciatic nerve gap	Enhanced nociceptive function, improved sciatic function index, and increased cross-sectional area of muscle	Increases downstream expression of NGF, promoting axonal outgrowth Enhances myelin debris clearance and remyelination
Valproic acid [123]	Mood stabilizer	Valproic-acid-loaded HAP/PDLLA/PRGD nerve conduit	10 mm rat sciatic nerve gap	Improved organization, density, and myelination of nerve fibers Improved nerve conduction velocity and compound muscle action potential	Increases SC metabolic activity and proliferation
Melatonin [124]	Sedative/hypnotic	3D melatonin/polycaprolactone nerve conduit	15 mm rat sciatic nerve gap	Improved functional recovery and electrophysiologic performance Enhanced mitochondrial activity	Antioxidant and anti-inflammatory effects at injury site Upregulation of cellular debris clearance
Simvastatin [125]	Lipid/cholesterol-lowering agent	Simvastatin in Pluronic F-127 hydrogel in chitosan conduit	10 mm rat sciatic nerve gap	Improved sciatic function index and electrophysiologic performance Increased number and diameter of myelinated axons, as well as thickness of myelin sheaths	Upregulation of endogenous neurotrophic factors: PTN, HGF, VEGF, and GDNF
4-aminopyridine [47]	Potassium channel blocker	4-AP in chitosan/HNT scaffold	15 mm rat sciatic nerve gap	Improved sciatic function index equivalent to autograft Increased myelin thickness equivalent to autograft	Increases BDNF and NGF release in SCs Upregulates release of myelin proteins in SCs
Methylcobalamin [126]	Active form of vitamin B12	MeCbl-loaded sheet and collagen sponge-filled PGA tube	10 mm rat sciatic nerve gap	Accelerated recovery of sensory function Increased myelinated axon area and count	Enhances axonal outgrowth, SC differentiation, and myelination Improves neuronal survival

Hsp90—heat shock protein 90; FKBP—FK506-binding protein; CyP40—cyclophilin 40; CsA—cyclosporine A; MP—methylprednisolone; Dex—dexamethasone; HAP—hydroxyapatite; PDLLA—poly D-L-lactic acid; PRGD—poly((lactic acid)-co-((glycolic acid)-alt-(L-lysine))); SC—Schwann cell; PTN—pleiotrophin; HGF—hepatocyte growth factor; VEGF—vascular endothelial growth factor; GDNF—glial-cell-line-derived neurotrophic factor; 4-AP; 4-aminopyridine; HNT—halloysite nanotubes; MeCbl—methylcobalamin.

**Table 2 bioengineering-11-00776-t002:** Commercially available nerve conduits and wraps.

Device Name	Year of Approval	Manufacturer	Form	Material	510 (k) Number
Fastube Nerve Regeneration Device	1985	Research Medical Inc., Salt Lake City, UT, USA	Not available	Not available	K850785
Neurotube	1999	NeuroRegen LLC, Bel Air, MD, USA	Tube	PGA	K983007
SaluBridge	2000	Salumedica LLC, Atlanta, GA, USA	Tube	PVA	K002098
Neuragen Nerve Guide	2001	Integra LifeSciences Corporation, Plainsboro, NJ, USA	Tube	Collagen, source not noted	K011168
NeuroMatrix	2001	Collagen Matrix Inc., Franklin Lakes, NJ, USA	Tube	Collagen from bovine tendon	K012814
Surgisis Nerve Cuff	2003	Cook Biotech Inc., West Lafayette, IN, USA	Tube	Porcine small intestinal submucosa	K031069
Neurolac Nerve Guide	2003, 2005, 2011	Polyganics BV, Groningen, The Netherlands	Tube	Poly(DL-lactide-co-ε-caprolactone)	K032115, K050573, K112267
NeuraWrap	2004	Integra LifeSciences Corporation, Plainsboro, NJ, USA	Tube with slit	Bovine collagen	K041620
NeuroMend	2006	Collagen Matrix Inc., Franklin Lakes, NJ, USA	Tube with slit	Bovine collagen	K060952
SaluTunnel	2010	Salumedica LLC., Atlanta, GA, USA	Tube with slit	PVA	K100382
CovaOrtho-Nerve Resorbable Collagen Membrane	2012	Biom’Up S.A.S., Saint-Priest, France	Wrap	Porcine collagen	K103081
AxoGuard Nerve Protector	2014	Cook Biotech Inc., West Lafayette, IN, USA	Wrap	Porcine SIS	K132660
NeuroFlex	2014	Collagen Matrix Inc., Oakland, NJ, USA	Tube	Collagen from bovine tendon	K131541
NeuraGen 3D Nerve Guide Matrix	2014, 2017	Integra LifeSciences Corporation, Plainsboro, NJ, USA	Tube with luminal filler	Bovine collagen/GAGs (chondroitin-6-sulfate)	K130557, K163457
Reaxon Plus	2015, 2018	Medovent GmbH, Mainz, Germany	Tube	Chitosan	K143711, K180222
Nerbridge	2016	Toyobo Co. Ltd., Osaka, Japan	Tube with luminal filler	PGA, collagen from porcine skin	K152967
AxoGuard Nerve Connector	2016	Cook Biotech Inc., West Lafayette, IN, USA	Tube	Porcine SIS	K162741
Reinforced Flexible Collagen Nerve Cuff	2017	Collagen Matrix Inc., Oakland, NJ, USA	Tube	Collagen from bovine tendon, absorbable polymeric suture filament	K170656
NeuroShield	2019	Monarch Bioimplants GmbH, Root, Switzerland	Wrap	Chitosan	K190246
VersaWrap Nerve Protector	2020, 2023	Alafair Biosciences Inc., Austin, TX, USA	Wrap	Calcium alginate and hyaluronic acid	K201631, K232029
NervAlign Nerve Cuff	2022	Renerve Ltd., Melbourne, Victoria, Australia	Wrap	Collagen from porcine pericardium	K202234
Nerve Tape	2022, 2024	BioCircuit Technologies Inc., Atlanta, GA, USA	Wrap with hooks	Collagen from porcine SIS, nitinol hooks	K210665, K233533
Axoguard HA+ Nerve Protector	2023	AxoGen Corporation, Alachua, FL, USA	Wrap	Porcine SIS, sodium hyaluronate, sodium alginate	K223640, K231708
Rebuilder Nerve Guidance Conduit	2024	CelestRay Biotech Company LLC, Bethesda, MD, USA	Tube	Poly(lactide-co-caprolactone), poly(lactic-co-glycolic acid), polylactic acid-b-polyethylene glycol	K230794

PGA—polyglycolic Acid; PVA—polyvinyl alcohol; SIS—small intestinal submucosa.

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
