# Peer review of "Biohacking Nerve Repair: Novel Biomaterials, Local Drug Delivery, Electrical Stimulation, and Allografts to Aid Surgical Repair"

_bioengineering, 2024, doi:10.3390/bioengineering11080776_

Round 1

Reviewer 1 Report

Comments and Suggestions for Authors

The review article titled, "BioHacking Nerve Repair: novel biomaterials, local drug delivery, electrical stimulation, and allografts to aid surgical repair" discusses the various interventions that are available to enhance the neural regeneration. The article has been very well written. The authors have drafted the manuscript in a detailed, yet easily followable manner. They have covered various concepts such as delivery of therapeutics, advancements in grafts etc. This manuscript would be interesting for wide readers such as physicians and researchers working on designing materials for nerve regeneration. The manuscript can be accepted for publication in its current form.

Author Response

Comment 1: 

  • The review article titled, "BioHacking Nerve Repair: novel biomaterials, local drug delivery, electrical stimulation, and allografts to aid surgical repair" discusses the various interventions that are available to enhance the neural regeneration. The article has been very well written. The authors have drafted the manuscript in a detailed, yet easily followable manner. They have covered various concepts such as delivery of therapeutics, advancements in grafts etc. This manuscript would be interesting for wide readers such as physicians and researchers working on designing materials for nerve regeneration. The manuscript can be accepted for publication in its current form.

Response 1: 

  • We thank reviewer one for their time evaluating our review and positive feedback. We hope that our additional revisions are accepted in a similar light, and appreciate any additional comments should they find them necessary.

Reviewer 2 Report

Comments and Suggestions for Authors

See File.

Author Response

Comment 1:

  • Some References are in all capitals.

Response 1:

  • We appreciate reviewer one’s attention to detail in all aspects of evaluation of our manuscript. The references in question have been updated on the newest draft of this review to avoid deviations from standard formatting.

Comment 2:

  • Tables: Can the authors or the editor reformat the 2 tables? They are hard to read. Suggestions are
    • Put lines between the columns and rows
    • Use single line spacing instead of double.
    • Make the reference numbers the same size font as the text
    • Don’t use full justification, because it results in words spread out and therefore can be confusing to figure out which column they belong to.

Response 2:

  • We thank reviewer two for these recommendations. Tables 1 and 2 have undergone reformatting, including removal of any spacing before and after a paragraph, single-spacing, left-alignment, and the addition of borders between cells. Reference numbers have been resized to match the text size used in the remainder of the table.

Comment 3: 

  • In Table 1, the distinction between the columns “Observed Effects” and “Proposed Benefits” is not clear. How is “Interaction with HSP90 through FKBP-52” a proposed benefit? If it is an observation, put it in that column. If this suggests something they found that is proposed to be a benefit, but perhaps was just hinted at and needs more work to establish, then I suggest clarifying the benefit. As far as I can tell, the two columns both describe observed effects.

Response 3:

  • We thank reviewer two for this comment. Our intent in using two distinct columns for this information was to separate those effects reported within each study model from those representing more overarching or generalized mechanisms of benefit. We have updated the header of this column to read “Proposed mechanism of benefit” and have updated the preceding column to read “Study results” to better portray this.

Reviewer 3 Report

Comments and Suggestions for Authors

This is a review on novel materials, local drug delivery, electrical stimulation and allografts to aid surgical repair after peripheral nerve injury. Peripheral nerve injuries, especially transection injuries, often result in incomplete and poor functional outcome. Therefore, the field of interventions supporting and improving regenerative capacities is still very important. The manuscript is well written and clinically important. I did not found major considerations.

Minor:

- Table 1: In each line please cite study only once in first raw.

Author Response

Comment 1:

  • Table 1: In each line please cite study only once in first raw.

Response 1:

  • We thank Reviewer three for their time in evaluating our review and their feedback. Table 1 has undergone additional reformatting, including the removal of duplicate references within table rows, as well as other formatting to ensure the table is easier to read and interpret.